

# Precursors and pathways: Dynamically informed extreme event forecasting demonstrated on the historic Emilia-Romagna 2023 flood

Joshua Dorrington[1], Marta Wenta[1], Federico Grazzini[2,3], Linus Magnusson[4], Frederic Vitart[4], and Christian M. Grams[1,5]

[1]Institute of Meteorology and Climate Research (IMKTRO), Department Troposphere Research, Karlsruhe Institute of Technology (KIT), Karlsruhe, Germany
[2]ARPAE-SIMC, Regione Emilia-Romagna, Bologna, Italy
[3]Ludwig-Maximilians-Universität, Meteorologisches Institut, München, Germany
[3]ECMWF, Shinfield Park, Reading, RG2 9AX, United Kingdom
[5]Federal Office of Meteorology and Climatology, MeteoSwiss, Zurich-Flughafen, Switzerland

**Correspondence:** Joshua Dorrington (joshua.dorrington@kit.edu)

**Abstract.** The ever-increasing complexity and data volumes of numerical weather prediction demands innovations in the analysis and synthesis of operational forecast data. Here we show how dynamical thinking can offer directly applicable forecast information, taking as a case study the extreme north Italian flooding of May 2023. We compare this event with historical north Italian rainfall events – in order to determine a) why it was so extreme, b) how well it was predicted, and c) how we may improve
our predictions of such extremes. Lagrangian analysis shows, in line with previous work, that extreme rainfall in Italy can be caused by moist air masses originating from the North Atlantic, North Africa, and, to a lesser extent, Eastern Europe, with compounding moisture contributions from all three regions driving the May 2023 event. We identify the large-scale precursors of typical north Italian rainfall extremes based on geopotential height and integrated vapour transport fields. We show in ECMWF operational forecasts that a precursor perspective was able to identify the growing possibility of the Emilia-Romagna
extreme event eight days beforehand – four days earlier than the direct precipitation forecast. Such dynamical precursors prove well-suited for identifying and interpreting predictability barriers, and could help build forecaster's understanding of unfolding extreme scenarios in the medium-range. We close by discussing the broader implications and operational potential of dynamically-rooted metrics for understanding and predicting extreme events, both in retrospect and in real-time.

## 1 Introduction

In Spring 2023, sustained heavy rainfall in northern Italy led to extensive flooding. Following earlier heavy rainfall between the 1st and 3rd of May, a second, 40-hour period of extreme precipitation in the Emiglia-Romagna between the 15th-17th of May caused more than 65,000 landslides (Brath, 2023) and breached rivers in 23 separate points, flooding 540 sq. km of lowland. Despite timely meteo-hydrological 'red' warnings issued 48 hours ahead – which triggered school closures and cancellation of transport – 16 casualties were unfortunately reported, as were over 9 billion euros of damages (ARPAE, 2023; AST, 2023).





The severity of the May 15th-17th rainfall (hereafter the 'case study event') was unprecedented in the 102-year instrumental history of Romagna. While the compound nature of the event – amplified by the pre-saturated soils from the May 1st-3rd rainfall – complicates assessment of return times, estimates range from 60 years to more than 500 years for individual basins (Brath, 2023), and 200 years for the 1st-to-18th May period which envelops both events (Barnes et al., 2023).

Current numerical weather predictions are generally able to make deterministic predictions of significant rainfall occurrence
1-3 days ahead, with an accurate prediction of intensity (relative to a model climate) perhaps 24 hours ahead (Haiden and Janousek, 2023). Probabilistic predictions of increased rainfall probability rarely surpass a week (Gascón et al., 2023; Leon, 2023). The case study event was therefore typical in this respect, despite its extreme nature, with skilful predictions of event occurrence approximately three days ahead in the ECMWF (European Centre for Medium Range Weather Forecasts) forecast (shown in Section 6). However, as early as the 8th May – 8 days before the event – several of the authors of this study
received early signs that such an extreme northern Italy rainfall event may occur, when no extreme signal was yet visible in precipitation forecasts. This signal came from an experimental monitoring of the large-scale circulation patterns that often lead to extreme rainfall events in different European regions within the ECMWF forecast, following an approach recently described in Dorrington et al. (2023).

In this paper, we will use this recent case study event as a demonstrative example to discuss how and why such early signs of
extreme events can be identified, and use Lagrangian trajectory analysis to delve deeper into the moisture sources and smaller-scale dynamical drivers of north Italian extreme precipitation, with reference to a climatology of such springtime precipitation extremes. We will also show that precursors can be used to understand predictability barriers in a simple way, amenable to operational use. In doing so we aim to demonstrate the potential insights and practical predictive value that dynamically-based perspectives can add to the toolbox of forecasters, and to researchers of predictability and extreme-events.

The specific lead-times at which European rainfall can be accurately forecast varies between regions, seasons and individual events, with mesoscale and convectively driven rainfall less predictable than frontal systems (Keil et al., 2014; Grazzini et al., 2021). However, in the mid-latitudes even mesoscale rainfall events are conditioned on the surrounding larger-scale circulation, with their occurrence encouraged or suppressed by Rossby wave activity, blocking events, and jet dynamics on the scale of thousands of kilometers. These large-scale 'precursors' are more predictable than the rainfall itself, and their modulating effect
on rainfall is not always well captured in models due to deficiencies in boundary-layer processes, orographic representation and precipitation parameterizations. Dorrington et al. (2023) recently showed in reanalysis data that national-scale extreme rainfall events, can be predicted to at least some degree across Europe in all seasons, by monitoring their typical large-scale flow characteristics, at timescales of 3-6 days ahead. In a numerical forecasting context, this should be thought of as a possible 'boost' of skill – if large-scale precursors can be predicted skilfully at day 5, then the resulting impact on rainfall may be
improved out to day 8-11, as has recently been shown in Grazzini et al. (2024). In addition to the possibility of quantitative skill improvements, there is also narrative value in analysing precursor activity, which could help the operational forecaster understand the physical basis of emerging forecast scenarios, and so better leverage their experience and intuition.

We briefly summarise the key dynamics of north Italian rainfall in section 2, introduce data and methodology in section 3, discuss the climatology of springtime northern Italy extreme rainfall in section 4, analyse the dynamics of the case study event





in section 5, and finally discuss forecast early warnings and predictability barriers in section 6. We finalise in section 7 by summarising our findings and discussing the broader relevance of our results for extreme event prediction, and present ideas for how to operationalise dynamical knowledge.

## 2   Dynamics of North Italian rainfall

On the largest scales a Rossby wave-dominated upper-level flow over the Euro-Atlantic, and eastward propagating Rossby
wave packets originating from North America, increase the probability of extreme rainfall in northern Italy in the following days (Grazzini et al., 2021). Directly, this is due to the development of an upper-level trough over the western Mediterranean favoured by wavy upstream flows, which drive strong southerly flow into northern Italy (e.g. Martius et al., 2008; Massacand and Wernli, 1998; Raveh-Rubin and Wernli, 2015). Grazzini et al. (2020) established elevated IVTmag (integrated water vapor transport magnitude) as a further key driver of heavy precipitation in the region.

However, large-scale features only set the stage for an extreme event, and smaller synoptic scale processes are responsible for directly causing convection. The presence of an upper-level trough supports the transport of warm, southerly airflow across the western Mediterranean basin (Martius et al., 2008; Massacand and Wernli, 1998), fueling the development of warm conveyor belts. These ascending airmasses are associated with Mediterranean cyclogenesis (Raveh-Rubin and Wernli, 2015; Dayan et al., 2015). In springtime the increased meridional temperature gradient along North Africa's coast and weak static
stability near the Atlas Mountains particularly favor the formation of cyclones over north Africa (Trigo et al., 2002). Raveh-Rubin and Wernli (2015) also found that in most of the analyzed cases, heavy precipitation events in the western, central, and eastern Mediterranean regions are often preceded by high local moisture anomalies over the Mediterranean Sea. Moisture uptake typically occurs within two days before a precipitation event, but in some cases, it can extend beyond five to ten days, particularly when remote moisture sources (e.g. North Atlantic) are involved.

The Mediterranean sea is a critical source of moisture for south-west European precipitation as revealed by Lagrangian moisture source diagnostics targeted on Italy (Drumond et al., 2011) and France (Duffourg and Ducrocq, 2011), but often requires additional moisture sources to intensify rainfall to extreme levels (Winschall et al., 2014). Duffourg and Ducrocq (2011) notes a flow-dependence, with local Mediterranean moisture sources the predominant driver of extreme rainfall during upstream anticyclonic conditions.

Studies using Lagrangian moisture diagnostics reveal the Mediterranean Sea as a critical moisture contributor for the extreme rainfall events in the western part of the basin. Drumond et al. (2011) in a comprehensive five-year study, identified the Mediterranean Basin as the central moisture region influencing extreme rainfall on the Italian Peninsula. Winschall et al. (2014) further expanded on this, finding that the Mediterranean Sea's surface is one of several important moisture sources for extreme rainfall events in the northwestern Mediterranean basin and additional sources are frequently required to intensify
these events to extreme levels.

Those conclusions are further reinforced by the analysis of a severe rainfall event in Southeast France (Duffourg and Ducrocq, 2011). Their findings align with those of Winschall et al. (2014), indicating that moisture for intense rainfall is



sourced from the Mediterranean Sea's evaporation shortly before the event and from distant sources over an extended period. Notably, they determine that the Mediterranean Sea's evaporation becomes the predominant moisture source during anticyclonic conditions in the days preceding the rainfall.

In order to study both the large-scale and synoptic factors that led to the case study event, we use both flow precursor and Lagrangian trajectory diagnostics. We identify the sources of the moisture that contributed to the rainfall and why case-study precipitation totals were so high, the role the large-scale flow played in supporting and conditioning for the event, and the added-value provided by large-scale precursors over the direct precipitation forecast.

## 3  Data and Methods

### 3.1  Data

The ERA5 reanalysis (Hersbach et al., 2020) is used over the period 1979-2021 (for climatological calculations) + May 2023 (for case study analysis) to study the large-scale flow associated with Italian extreme rainfall events. For precursor analysis, geopotential height at 500hPa (Z500), surface pressure (SP), zonal and meridional 850hPa wind (U/V850) and the magnitude of vertically integrated water vapour transport (IVTmag) were all used at 12-hourly, 1 degree resolution. ERA5 total precipitation was downloaded at 0.25 degrees. When anomalies are plotted, the fields have been deseasonalised using a day-of-year seasonal cycle smoothed with a 30-day running mean. For the calculation of Lagrangian trajectories (Section 3.3) we use the full 3D fields of ERA5 on model levels interpolated on a horizontal latitutude-longitude grid with $0.5° \times 0.5°$ resolution and at 3 hourly temporal resolution (also in the period 1979-2021 for climatological calculations and May 2023 for the case study). In order to more confidently identify the exact times and locations of most intense rainfall during the May 2023 Emilia-Romagna case, we use half-hourly IMERG data, an integration of multi-satellite rainfall observations, at 0.1 degree resolution (Huffman et al., 2019) and the ARCIS gridded high-resolution rain gauge data (Pavan et al., 2019). To study predictability, we analyse the evolution of the May 2023 case study in 100-member ECMWF extended range forecasts from the model cycle 48r1 (Haiden and Janousek, 2023), which was running daily in pre-operational mode at the time. We consider starting dates between the 5th May and the 16th May, and download total precipitation, Z500 and IVTmag fields at 0.5 degrees, sampled down from the native horizontal resolution of $\approx$ 36km. In order to compute precursor indices in forecast data (see below) we compute deseasonalised Z500 and IVTmag anomalies by removing a centred 3-week leadtime-dependent climatology, computed using 11-member hindcasts covering the period 2003-2022 from the same model cycle.

### 3.2  Rainfall Precursor Indices

Precursors of North Italian heavy rainfall events are identified following exactly the method recently presented in Dorrington et al. (2023) Section 3.3, which we now summarise. We begin by defining a Boolean time series of heavy rainfall events. This is done by spatially averaging MAM 48 hourly total precipitation from ERA5 over the domain of northern Italy (mask shown in Figure S1), and defining an event as a day with rainfall exceeding the 90th percentile of this index ($\approx$ 8.5mm/day).





We then compute lagged anomaly composites of deseasonalised large-scale field anomalies in the days preceding these heavy rainfall events. These composites are masked to focus on the regions of key activity: gridpoint anomalies that are statistically insignificant (p>0.05) or have an amplitude $< 0.25\times$ the gridpoint climatological standard deviation are masked. Groups of unmasked gridpoints that comprise a connected area of less than 250,000 km2 (i.e. equivalent to 20 1x1 deg equatorial gridpoints) were also masked. Finally, all unmasked gridpoints were convolved with a 5-point lat-lon square: all gridpoints within 2 degrees of an unmasked gridpoint were unmasked. This process produces smooth, large-scale, statistically robust patterns that are associated with increasing the likelihood of extreme rainfall. Finally, these patterns are used to produce precursor indices: each pattern is projected onto the corresponding large-scale deseasonalised-anomaly field to produce a scalar index, which is standardised, by subtracting the mean and dividing by the standard deviation, both estimated based on a 1979-2021 MAM ERA5 climatology. Positive index values indicate increased likelihood of an extreme, while negative values indicate decreased likelihood. To produce IFS forecasts of precursor activity, the deseasonalised forecast fields are projected on the corresponding ERA5 precursor patterns, and approximately standardised using the ERA5 index mean and standard deviation.

### 3.3 Lagrangian analysis

We use the LAGRANTO (Sprenger and Wernli, 2015) analysis tool to derive kinematic Lagrangian trajectories based on three-dimensional winds from the ERA5 dataset (Hersbach et al., 2020) of higher temporal and horizontal resolution. The positions of these trajectories are updated every 3 hours, and along these trajectories, we trace parameters such as: pressure height ($p$), temperature ($T$), specific humidity ($Q$), relative humidity ($RH$), potential temperature ($\theta$), surface latent heat flux ($SLHF$), surface sensible heat flux ($SSHF$), 2m air temperature ($T2M$), and underlying sea surface temperature ($SST$) and boundary layer height ($BLH$).

For our designated study region in northern Italy, the starting points of trajectories are identified using the methodology of Sodemann et al. (2008). We establish an equidistant grid of 50x50km, framed by latitudes 42 to 47°N and longitudes 7 to 15°E (c.f. Figure S1). On the vertical scale, these points range from 1000 to 480 hPa, with intervals set by $\Delta$p=25 hPa, considering only starting points that have a relative humidity exceeding 80% and that are over land. Trajectories are initiated every 3 hours, starting from a specified time of extreme rainfall event and extending 48 hours backward. Each trajectory is run for a duration of 240 hours backward and 12 hours forward from its starting point, defined as time 0.

Based on the characteristics and directions of the trajectories, they are classified into the following distinct categories:

- North African 'Low' Trajectories (NAlow): These trajectories travel south of latitude 36°N within 168 hours (7 days) before their initiation and, when positioned there they consistently register a pressure exceeding 800 hPa.

- North African 'Up' Trajectories (NAup): Similar to NAlow, these are located south of latitude 36°N within the -168 hours (7 days) preceding their initiation. Upon reaching this latitude, they register a pressure smaller than 800 hPa. Considering that those trajectories represent only a small fraction of all started trajectories ($\sim$5.4%) and do not contribute significantly to precipitation in any of the analyzed events, they have been excluded from further analysis.





- **East Trajectories (EAST):** This group includes trajectories that, -120 hours (-5 days) before arrival into the starting region are located east of longitude 15°E. It excludes trajectories classified as NAlow of NAup.

- **West Trajectories (WEST):** This category encompasses all trajectories, that -120 hours (-5 days) before arrival into the starting region are located west of longitude 15°E and are not classified as EAST, NAlow, or NAup.

On average 0.13% of trajectories do not fit into any of the categories described above. We consider data from 5 and 7 days prior for the identification of NA and EAST/WEST trajectories, respectively. This approach is taken because NA trajectories typically remain in the Mediterranean region for up to 5 days after leaving North Africa and before contributing to rainfall.

### 3.3.1 Identification of Moisture Origins and Loss

The origins of moisture are identified following Sodemann et al. (2008), which has been widely adopted (e.g. Papritz et al., 2021; Xin et al., 2022; Jullien et al., 2020; Aemisegger and Papritz, 2018; Winschall et al., 2014). In this technique, moisture uptakes (i.e. specific humidity gains) along a given trajectory are defined as any 3-hourly specific humidity increment exceeding 0.02 g/kg ($q_{t+3} - qt > 0.02$). Each such uptake is allocated a relative weight based on the subsequent humidity shifts within that trajectory. Thus, the overall influence of each uptake is re-calibrated by taking into account any precipitation events and subsequent uptakes it encounters.

Overall, our analysis indicates that the method used for identifying moisture uptakes explains about 80% of the moisture content within the studied trajectories. The remaining 20% may be attributed to processes occurring above the boundary layer, such as advection, horizontal turbulent mixing, or the evaporation of precipitation.

Similarly, moisture losses (i.e. specific humidity losses) along a trajectory are defined as negative 3-hourly specific humidity tendencies ($q_t - q_{t+3h} > 0$). Following the approach of Sodemann et al. (2008) we determine that negative specific humidity tendency indicates precipitation. We consider only those losses of moisture that occurred within the North Italian starting region of trajectories (42° to 47°N and 7° to 15°E) and within the preceding 48h from trajectory start.

## 4 Climatological perspectives on extreme rainfall in northern Italy

### 4.1 Moisture sources of extreme rainfall

We begin by characterising the dynamics of MAM North Italian rainfall extremes from a Lagrangian perspective, identifying the sources of moisture and the properties of the contributing moist air masses in the preceding 10 days. We consider the 100 MAM events with the highest 48h rainfall accumulations over northern Italy as identified in ERA5 between 1979-2021. We then discard 34 events that occur within 24 hours of another, stronger, event, leaving 66 independent 2-day rainfall extremes. These are then a subset of the upper decile rainfall events used for defining precursor patterns, which benefits from a larger number of samples.

Considering the trajectories of precipitating airmasses over our climatology, we identify three distinct pathways, shown in Fig. 1a). Defined quantitatively in Section 3.3, these pathways include trajectories passing over North Africa in the lower



troposphere (NALow), those originating east of the starting region (EAST), and those coming directly from the North Atlantic (WEST). Each pathway is unique, ensuring that trajectories classified in one category are not found in the others.

The sources of moisture uptakes for all the identified airmass pathways include both land and marine regions, and are all relatively local, in agreement with Winschall et al. (2014). Individual moisture uptake locations are weighted by their contribution to total moisture content prior to precipitation ($f$), ensuring that uptake density remains unaffected by the density of trajectories (Fig. 1). NALow trajectories primarily source moisture from the coastal areas of North Africa and the central Mediterranean, with additional, smaller, moisture contributions from the Iberian Peninsula and Italy (Fig. 1b). The moisture

uptake region for these trajectories is widespread, with the densest uptakes occurring along the Tunisian coast and Tyrrhenian Sea. WEST trajectories also gather a significant amount of moisture from the Mediterranean Sea's surface. However, northern Italy itself proves to be the most significant uptake region, highlighting the significance of local moisture recirculation in contributing to extreme rainfall events along this pathway. Additional non-local land sources of lower significance for WEST trajectories include the Iberian Peninsula and southwestern France (Fig. 1c). EAST trajectories (Fig.1d) have comparatively

low moisture uptakes, as they overall contribute less rainfall to the analysed events. The majority of moisture carried by those air masses is sourced from the Adriatic Sea and the coasts of eastern Italy and Croatia.

Since the Mediterranean sea has experienced an exceptionally long marine heatwave event starting in 2022 and lasting throughout 2023 (Marullo et al., 2023), coinciding with our case study, we considered whether SST anomalies systematically increase moisture uptake intensity. While the majority of moisture uptakes do occur over the ocean, the correlation between

low-level moisture uptakes and underlying SST anomalies in MAM is in fact fairly weak (c.f. Figure S2), suggesting SSTs are not necessarily a source of predictive skill for rainfall. What can be inferred from these results is that the most intensive uptakes for all identified pathways occur over warmer waters (290-295 K), indicating that SSTs set an upper bound on uptake strength, but do not guarantee strong uptake in isolation. Of course it is likely that SST anomalies may be more predictive during the warmer ocean periods JJA and SON (Sanchez et al., 2023).

NALow trajectories contribute the greatest fraction of precipitation, followed by WEST trajectories (Fig. 2). As we will see, these pathways can be broadly linked to cyclogenesis in the lee of the Atlas mountains or Alps respectively. In contrast, and as mentioned, the EAST trajectories generally play a minor role in northern Italy's extreme rainfall events but contribute substantially in a few cases – including the case study event. The rainfall totals assigned to each pathway for each event are essentially uncorrelated (corr<0.1 in all cases), indicating that these spatially separated trajectories are also dynamically

independent: the presence of moist air masses from one source does not support or suppress moisture influx from another source.

Each trajectory pathway also has distinct dynamical properties, which modulate their ability to hold or precipitate moisture. Within the analyzed trajectory subsets, NALow trajectories are warmest (Fig. 3b), as they spend several days in the lower latitudes (<30°), particularly over North Africa (c.f. Fig 1a). They also display the highest potential and equivalent potential

temperatures (Fig. 3e,f). Their high thermal energy allows them to hold large volumes of moisture (Fig. 3c) and decreases their stability allowing them to rapidly ascend (Fig. 3a). They therefore exhibit the largest negative specific humidity tendencies (i.e. produce the most rainfall) compared to WEST and EAST trajectories. Furthermore, these air masses may play a role in



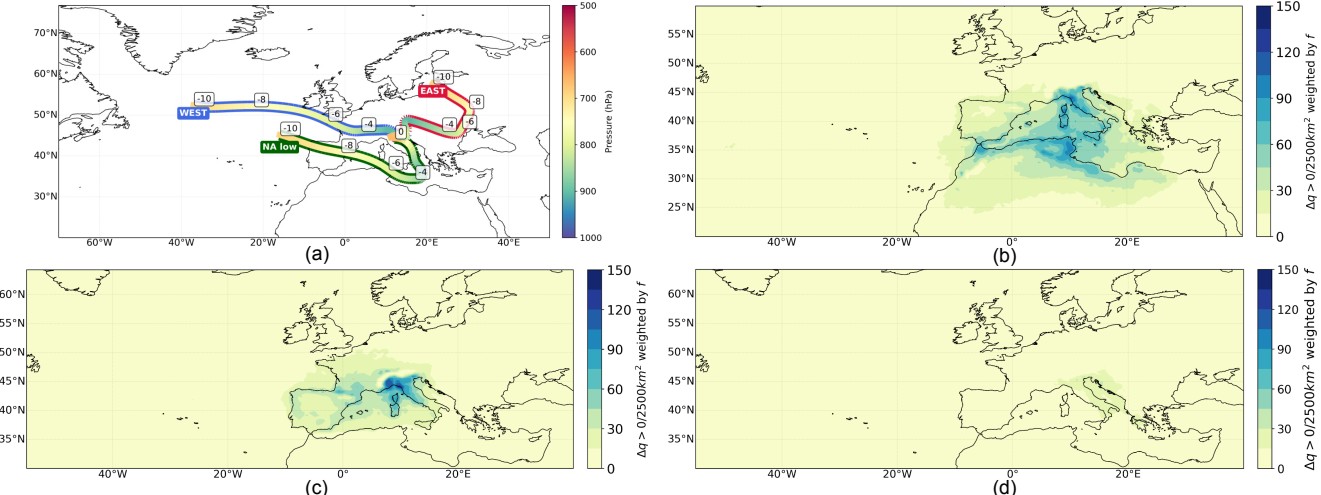

**Figure 1.** (a) Location and altitude of Lagrangian trajectories that contribute to MAM north Italian rainfall between 1979-2021, averaged over each pathway. Time prior to rainfall is indicated in days. The number of uptakes per 2500 km $^2$ ($\Delta q(q_{t+3h} - q_t) > 0$) weighted by the contribution of uptake to total moisture content before precipitation ($f$) is shown for (b) NAlow trajectories, (c) WEST trajectories, (d) EAST trajectories.

intensifying convective updrafts. An indicative sign of this is a slight drop in potential temperature prior to ascent (Fig.3f, time -20 to -10 h), suggesting unstable atmospheric conditions. While they have lower specific humidity than EAST trajectories (Fig. 3c), NAlow trajectories show a rapid increase in relative humidity as precipitation begins, aligning with their major role in the analyzed events. In comparison, the WEST and EAST trajectories, in terms of potential and equivalent potential temperatures (Fig. 3e,f), are similar but do not exhibit the same buoyancy and tendency for convection as seen in NAlow trajectories. Overall, EAST trajectories, are slightly warmer (Fig. 3b), more humid (Fig. 3c), and generally occur in lower layers of the atmosphere than WEST trajectories.

## 4.2 Dynamical precursors of extreme rainfall

The coherent Lagrangian pathways we have just identified are connected to robust dynamical flow precursors prior to north Italian extreme rainfall on large scales. Documented extensively in recent years (Dorrington et al., 2023; Grazzini et al., 2021) these precursors isolate configurations of the large-scale flow which favour the occurrence of a localised extreme in the following days. The details of the precursor circulations undergo seasonal fluctuations as the large-scale background state evolves, and while they are more clearly visible at long lead times during SON and DJF, they are also relevant in MAM and still robust at time lags of 0-2 days (Dorrington et al., 2023). Figure 4 shows lagged composite anomalies of large-scale fields in the days preceding MAM 90th percentile 48 hourly North Italian rainfall events, taken between 1979-2021, as described in Section 3. An Atlantic Rossby wave packet is visible in Z500, SP and V850 anomaly composites at a two-day lag prior to the events, with a strengthening western European trough leading to strong southerlies into Italy. Negative SP anomalies




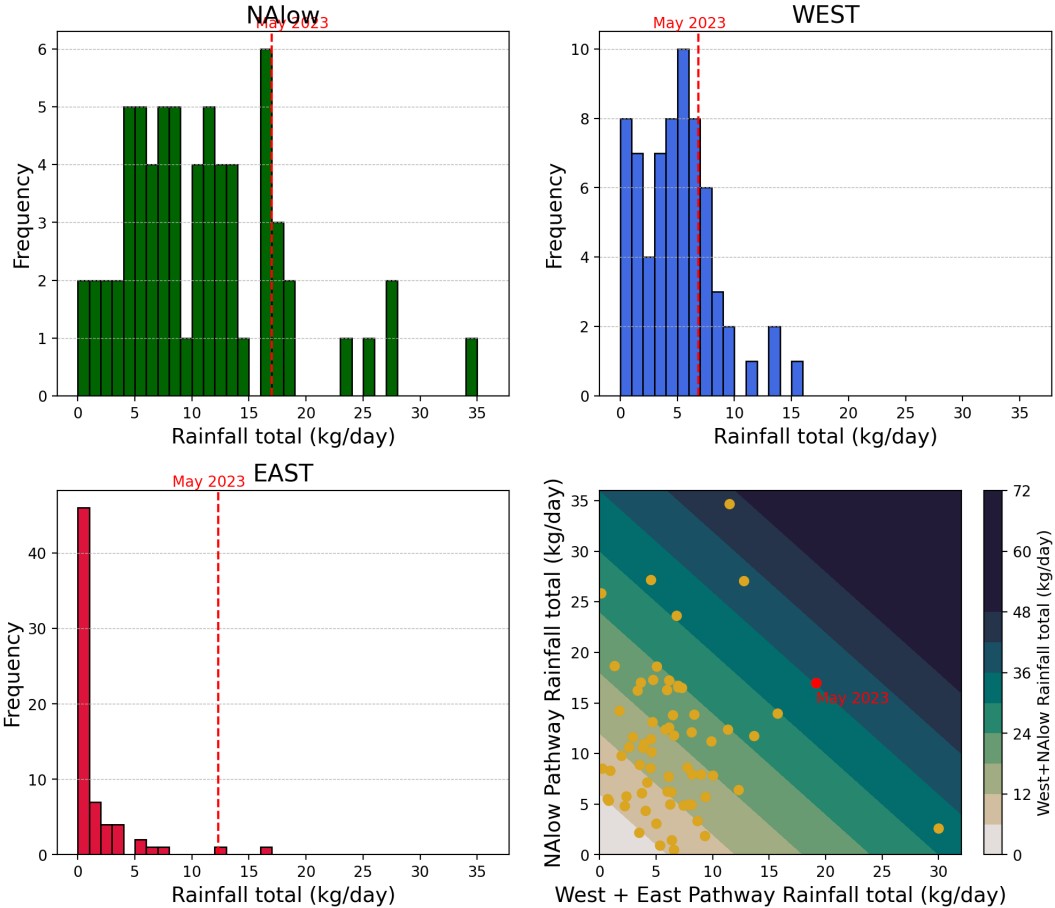

**Figure 2.** Histograms of total precipitation attributable to the (a) NAlow, (b) WEST, and (c) EAST pathways during 66 48 hour extreme rainfall events in northern Italy. The case study event, from 15th-17th May 2023 is shown with a red dashed line. (d) shows a scatter plot of WEST+EAST vs NALow rainfall totals, indicating their low correlation and that the case study featured high rainfall from all pathways.

are shifted slightly east compared to Z500 anomalies, indicating baroclinic tilt with height, and extend notably further south, coinciding with Westerly anomalies over the Atlas mountains, visible in U850, which drive lee cyclogenesis. Positive IVTmag anomalies over the Mediterranean jump sharply in amplitude on the day of the events, which is consistent with the localised moisture sources shown in figure 1b-d.

In Dorrington et al. (2023) it was proposed that lagged composites of typical flow precursors could be used to produce corre-
sponding precursor activity indices: measures of the projection of an instantaneous anomaly field onto the expected precursor flow field for a particular extreme event class. These standardised activity indices can then be used as scalar summaries of the circulation, with positive values indicating an increased event likelihood. The black contours in figure 4 indicate the regions of the composites used to compute such indices, following the filtering procedure described in section 3. Other regions of each





**Figure 3.** Properties of trajectories averaged over the pathways, showing pressure in hPa (a), temperature in Kelvin (b) specific humidity in grams/kilogram (c) relative humidity (d), equivalent potential temperature (e) and potential temperature (f) both in Kelvin. The solid lines represent the climatology of 66 extreme rainfall events, while the dashed lines denote the case study event of May 2023. The time on the x-axis indicates the trajectory duration, with 0 marking the start and -240 representing ten days prior.







**Figure 4.** Filled contours show lagged anomaly composites of upper decile MAM northern Italy rainfall events, 2, 1, and 0 days before precipitation. Black contours bound regions used to define precursor patterns: these regions have statistically significant anomalies relative to climatology, an amplitude >0.25 times the climatological standard deviation of the variable at each grid point, and represent coherent areas of >20 square degrees. See methods for a more detailed explanation.





composite are masked, and precursor activity indices are computed by projecting only onto the unmasked regions. Due to the
high correlation (>0.8) between many of these precursor activity indices, and for simplicity, we will here consider only the
Z500 lag 0 and IVTmag lag 0 indices, which capture projection onto a zonally-oriented ridge-trough dipole over Italy, and
a strong positive Mediterranean moisture transport anomaly with suppressed transport over the UK, respectively. Figure S4
shows example daily fields corresponding to positive and negative precursor activity in both cases, while Figure S5 confirms
that these scalar indices are well-correlated with actual rainfall totals over northern Italy. We will explore the use of these
indices in forecasting the May 2023 event in section 6.

However, before we do so, we wish to understand in more detail how the distinct uncorrelated moisture sources identified
using Lagrangian trajectories relate to this story of 'typical' flow precursors. To do so, we composite the 66 extreme events
for which trajectories were computed, dependent on their dominant moisture source. Specifically, we select events where the
precipitation from a given pathway was in the upper quintile of the 66 events, and where precipitation from the other two
pathways was not. This gives a clear separation between the pathways, and provides between 6-12 events for each class. Z500
composites are shown in Figure S6, but most clarity comes from composites of surface pressure and 850hPa wind, as shown in
figure 5.

Superficially, the composites share many similarities: a western anticyclonic anomaly, a European low-pressure anomaly
and a developing cyclonic system to the west of Italy. However, the details of the flows are quite distinct. NAlow-driven
events are associated with a strong Atlantic ridge and a tilted low over Scandinavia and the UK, which favours a strong
'L' shaped low-level jet, with northerlies off Portugal and westerlies over the Atlas mountains, with clear similarities to the
precursor composites of figure 4. The developing lee cyclone then tracks north-east into northern Italy, embedded in an overall
environment of high pressure. East-driven events feature less jet involvement due to anticyclonic anomalies prevailing over
Europe which deflect the jet north. The near-cutoff low pressure system which develops is shifted south and develops rapidly.
Finally, for 'West'-driven events the anticyclonic anomaly is shifted east, producing neutral conditions over North Africa and
guiding the jet north of the Pyrenees and directly into the western Mediterranean, where cyclogenesis occurs in the lee of the
Alps.

Clearly, the composites shown in figure 4 blur these different scenarios to an extent, and represent only an approximate mean
picture of the circulation prior to extremes. However, as we will see below, at least for the May 2023 case, which was unusual
in several ways, this simplification proves useful for understanding the dynamical flow precursors.

## 5   Dynamics of 15th-17th May 2023

We now move from the general to the specific, to understand how the climatological perspectives of our trajectories and
precursors correspond to an individual concrete case.

The left panel of figure 6 shows rainfall accumulations over the 40 hour period between 15-05-2023 18:00 and 17-05-2023
06:00 – our case study, discussed in the introduction. While extreme rainfall occurred in Sicily, and also along the Adriatic
coast and southern Bosnia, rainfall totals were particularly severe over Emilia-Romagna, whose position is approximated by



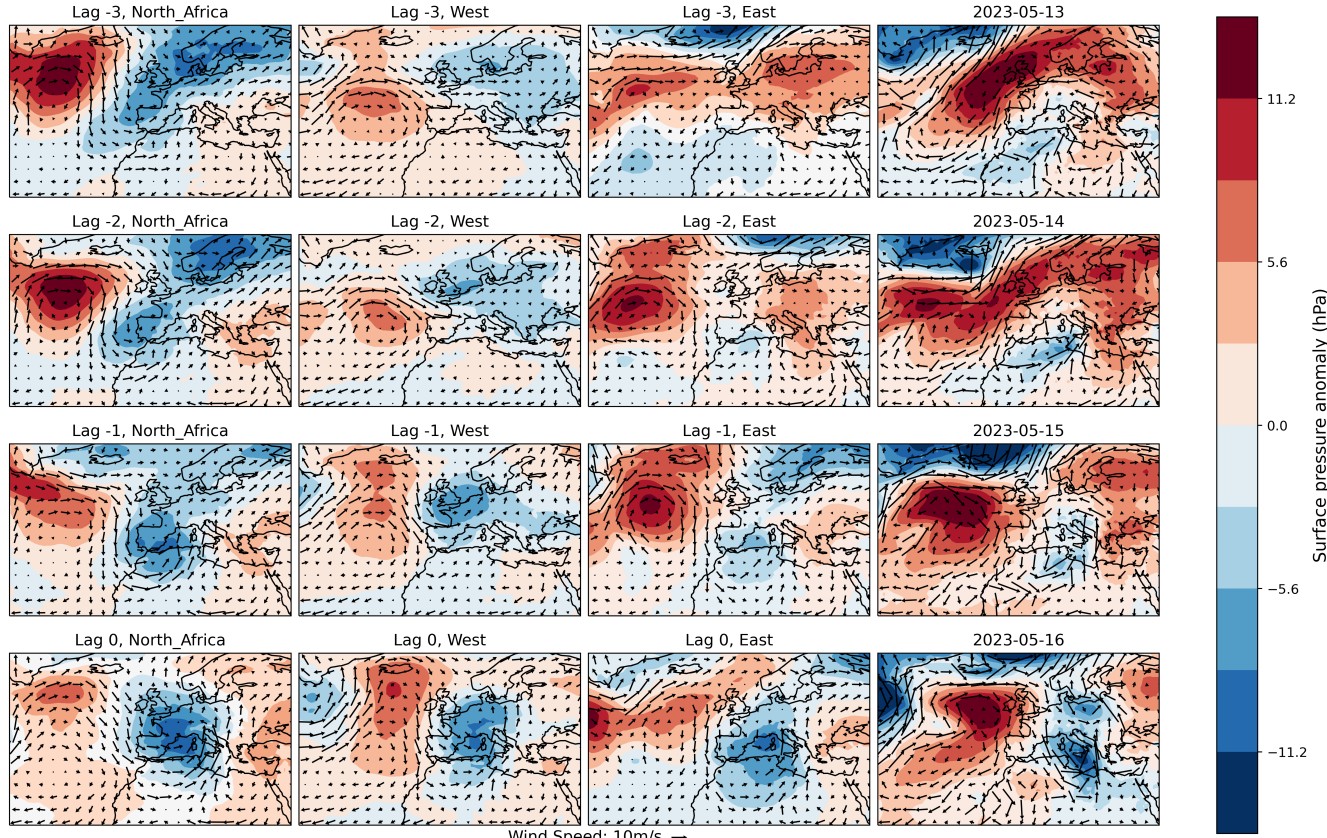

**Figure 5.** Lagged composites of surface pressure anomalies (filled contours) and full field 850hPa wind (arrows) 3-0 days before rainfall pathway composites. Each composite consists of a subset of rainfall events dominated by Lagrangian moisture transport from a specific source region (see main text for details). The right-hand column shows the case-study event, which received large moisture contributions from each source region.

the red box covering the domain [11-13E, 44-45.5N]. The right panel of figure 6 shows rainfall accumulated over the red box, indicating that this 40-hour period well isolates the event. The half-hourly IMERG satellite data indicates that there were two distinct peaks in precipitation, in the morning of the 16th and 17th respectively, and fully consistent with the daily ARCIS

gauge data. This particular period of rainfall was caused by storm Minerva, which tracked from the Southern Mediterranean over Sicily and into the Southern Tyrhennian Sea. Figure 7 provides a snapshot of the structure of Minerva during this first rainfall peak, with low surface pressure centred over the Appenines, a tilted upper-level trough and fast, moist Easterlies coming from the Adriatic into Emilia-Romagna.

However, it is cyclogenesis over the Labrador Sea, not the Mediterranean, which provides the first identifiable step in a

chain of dynamical events that ultimately led to the Emilia-Romagna flooding. To shed light on this downstream development from the Labrador sea towards the Mediterranean, Figure 8 shows full geopotential height fields from the 9th-17th May, with





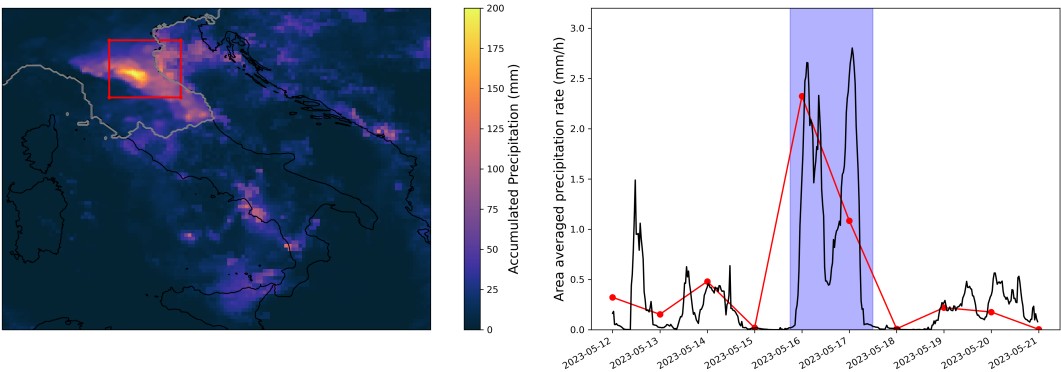

**Figure 6. Left**: observational rainfall accumulations over the 40-hour window from 15/05/2023 18:00 until 17/05/2023 12:00, based on ARCIS rain gauge data within the grey contour, and GPM-IMERG satellite data in the wider domain. The Emilia-Romagna region is marked with a red box. **Right**: In black: half-hourly precipitation rates from GPM IMERG, averaged over the Emilia-Romagna box, with the 40-hour heavy rainfall period shaded in blue. In red: daily ARCIS precipitation data averaged over the same Emilia-Romagna box.

anomaly fields shown in Figure S7. On the 10th, a pre-existing low over Newfoundland rapidly deepened and tracked east toward Greenland. Interacting with the pre-existing wavy Atlantic flow, this Labrador sea cyclone drove strong ridge building in the mid-Atlantic as seen on the 11th, and this ridge then broke anticyclonically on the 12th, producing a cutoff low pressure
system over south-western Europe and a strong anticyclonic anomaly over the Azores, inducing Northerly flow down through Iberia. The consequence was a strong 'L'-shaped low-level jet with strong low-level Westerlies over the Atlas mountains on the 14th (c.f. figure 5, right column), and rapid lee-cyclogenesis on the east side of the range. The remnant east-Atlantic ridge, widening and decaying by the 16th, also supports North-Westerly flow around the Alps and into Italy. At the same time, the anticyclonic anomaly over eastern Europe supports easterly flow into the central Mediterranean.

From a precursor index perspective, both Z500 and IVTmag activity indices were positive during the event, with IVTmag strongly so from the 14th onward; 2 standard deviations above normal (c.f. Figure 9). This envelope of large-scale precondi- tioning well captures the development, landfall and decay of Minerva. Interestingly, this was not the case for the earlier rainfall on the 10th: while IVTmag precursors are slightly positive, the Z500 precursor is 2 standard deviations below normal, which corresponds to the West European ridge and tilted Eastern Mediterranean low visible for the 10th in Figure 8. Such small-scale
cut-off lows are not well accounted for in the large-scale precursor framework, and this is partly responsible for the overall reduced coherence of flow precursors in MAM and JJA noted in Dorrington et al. (2023).

The spatial structure of Lagrangian moisture source trajectories for the case study closely resembles the climatological data (c.f. Fig. 1a and Fig. 10a). A key observation is that NAlow trajectories exhibit a distinct easterly turn, indicating recirculation over North Africa and the eastern Mediterranean. In contrast, EAST trajectories notably extend further eastward, reaching as
far as the Black Sea. NAlow trajectories predominantly absorb moisture from the eastern Mediterranean and near Italy (Fig. 10b). The most intense moisture uptakes, exceeding those in both WEST and EAST trajectories, occur over the warm (290-295



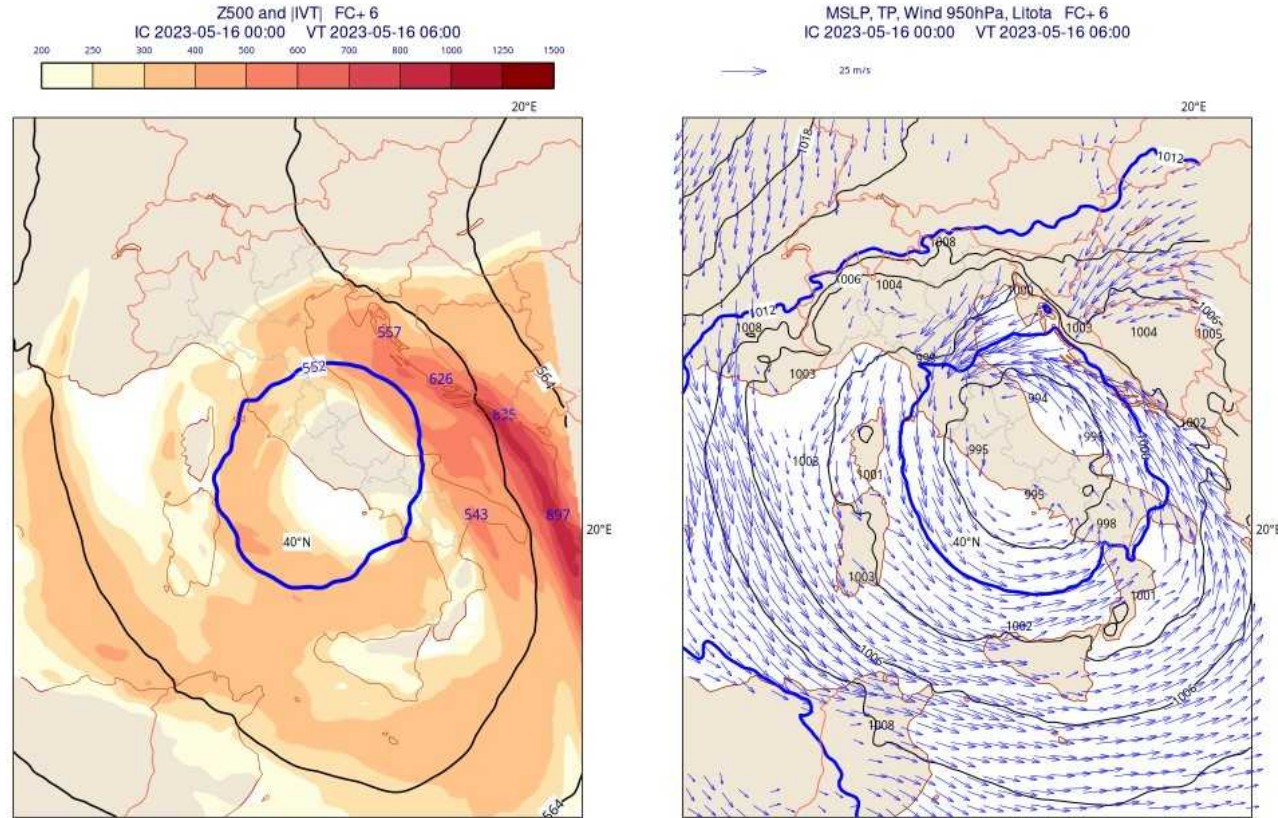

**Figure 7.** A representative snapshot of the synoptic circulation around Italy, as captured by the ECMWF high-resolution forecast, initialized 2023-05-16 at 00 UTC. Left: 500hPa Geopotential height (contours, dam) and the magnitude of integrated water vapour transport (IVTmag, kg*m/s) in shading. Right: Mean sea level pressure (black and blue contours, hPa) and 950 hPa wind (arrows, m/s). .



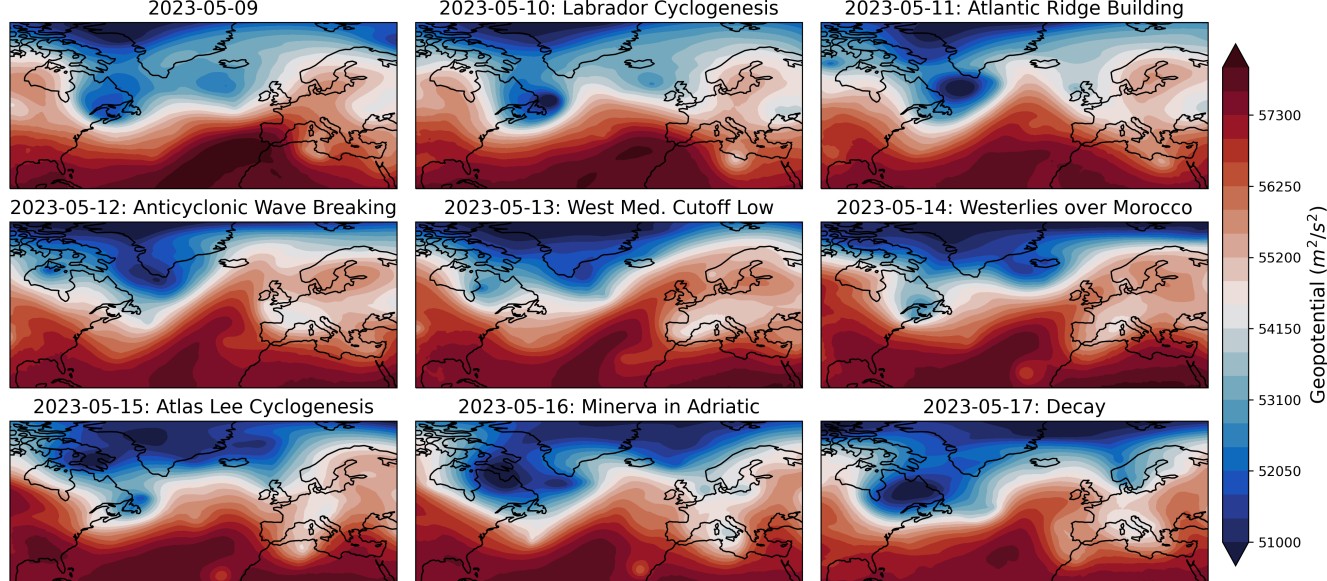

**Figure 8.** Euro-Atlantic circulation in the week preceding the flooding. Rapid cyclonic intensification over Eastern Canada and the Labrador Sea drives the development of a sharp Atlantic ridge, which breaks anticyclonically. This creates a cutoff-low over Western Europe, which propagates South and East, producing the Mediterranean cyclonic flow shown in figure 7. Equivalent eopotential anomalies are shown in Figure S7.

K) southern Mediterranean (c.f. Figure S2). WEST trajectories, originating from the North Atlantic, absorb moisture from both sea and land, particularly around the Emilia Romagna region (Fig. 10c). For EAST trajectories, most of the moisture uptakes occur through land-based evapotranspiration, particularly along the Croatian coast and from distant sources such as the Black Sea, (Fig. 10d). Furthermore, the moisture sources of EAST trajectories associated with warm sea surface are more intensive compared to those of WEST trajectories (Figure S2). This indicates that EAST trajectories might have greater capacity for transporting larger amounts of moisture. This is further supported by the subsequent analysis, which reveals that these trajectories maintain higher levels of equivalent potential temperature. (Fig.3e). In the comparison of this specific event with the climatology of 66 events from 1979-2021, we find that the moisture sources for NAlow and WEST trajectories are in fact similar to those observed historically (Figs. 10 and 1b-d). The moisture uptakes of NAlow air masses exhibit a slight eastward shift, aligning with the easterly turn depicted in Fig. 10a. However, it's the EAST trajectories that most notably diverge from the climatology (Figs. 10d and 1d), displaying a denser concentration of moisture sources. This characteristic is in line with their substantial contribution to the significant precipitation observed in May 2023.

The overall remarkably intense rainfall during the case study is due to strong rainfall from all three pathways including the uncommon EAST pathway (Fig. 2c). This confluence of different sources marks the event as particularly unique, and as a kind of compound extreme. This was the cause of the extended duration of the event, and the two rainfall peaks. As shown in Figure




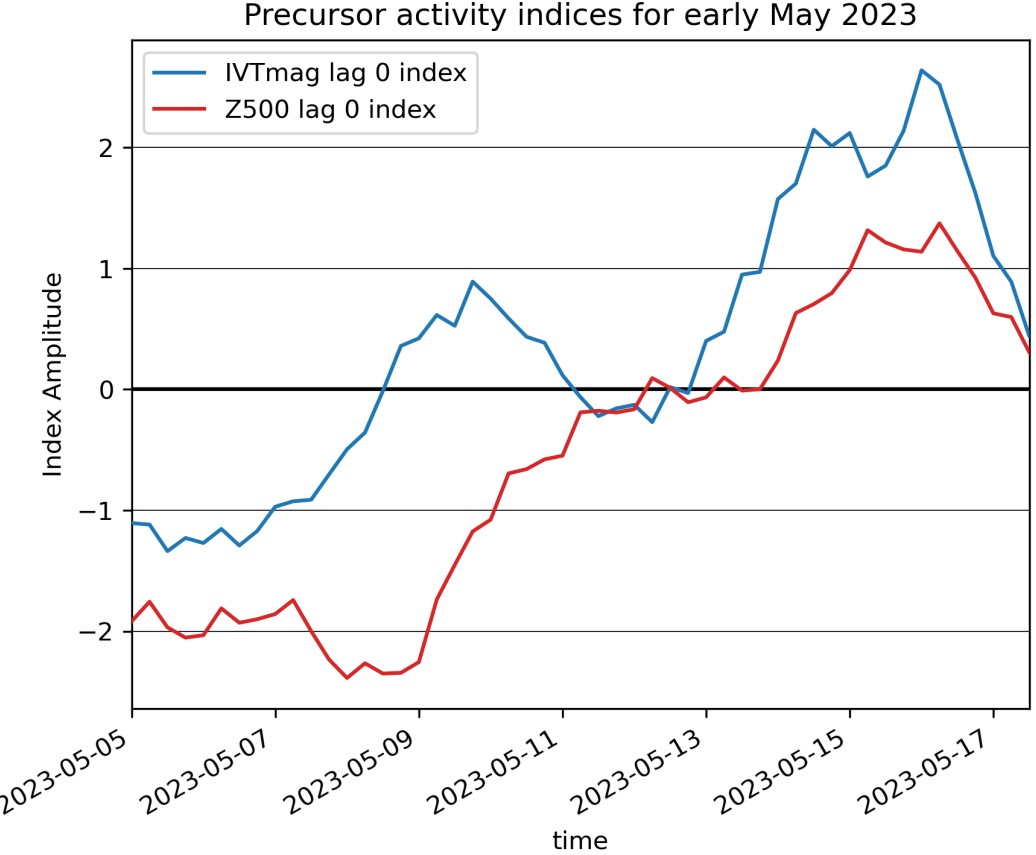

**Figure 9.** Evolution of IVTmag and Z500 lag 0 precursor indices, in the period before and during the case study event. Both indices are standardised to mean 0 and standard deviation 1, and represent the occurrence of strong vapour transport over the Mediterranean, and a trough-ridge dipole centred on Italy respectively (c.f. figure 4).

11 the rainfall on the 16th came primarily from NAlow airmasses accompanying Minerva, while the rainfall on the 17th was primarily due to later influx of moist westerly and easterly winds.

325   The reason for this confluence of trajectories can be understood by comparing the low-level flow anomalies of the May 2023 event to the pathway composites of figure 5. We see the flow anomalies for the 13th were most similar to the EAST-driven events with strong and persistent anticyclonic anomalies in eastern Europe, but with the important difference that the wave breaking has produced a true cut-off low over the western Mediterranean. This produces strong enough westerlies over the Atlas to trigger cyclogenesis, even without an Iberian low, as is typical in the NAlow driven events. Finally, the breakdown of the wave ultimately produces a strong anticyclonic anomaly on the 15th in just the right location to support north-westerlies

330   over France, as in the WEST pathway. As such, airmasses from all three pathways are able to converge into northern Italy over the 40 hour period of the event.





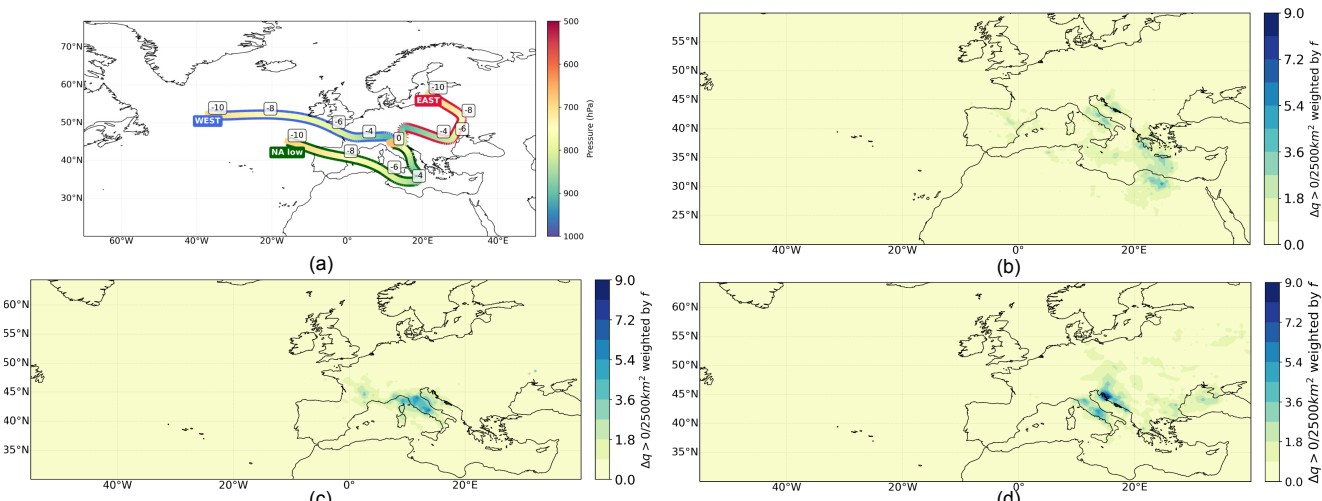

**Figure 10.** (a) Location and altitude of Lagrangian trajectories that contribute to north Italian rainfall in the case study event of 15-17 May 2023, averaged over each pathway. Time prior to rainfall is indicated in days. Moisture uptake ($\Delta q(q_{t+3h} - q_t) > 0$) density per 2500 km$^2$ weighted by the contribution of uptake to total moisture content before precipitation ($f$) is shown for (b) NAlow trajectories, (c) WEST trajectories, (d) EAST trajectories.

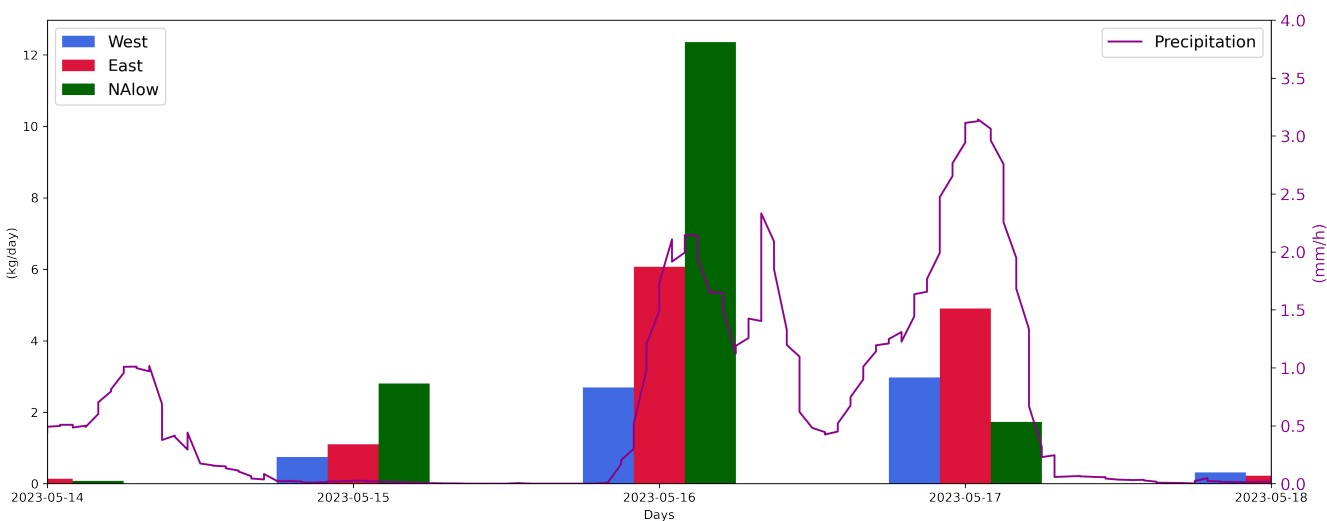

**Figure 11.** Total moisture loss ($\Delta q(q_{t+3h} - q_t) < 0$) and precipitation ($mm/h$) during the case-study event. The bars, divided into pathways from Fig.10, represent the total moisture loss of trajectories in kg/day for the dates specified on the x-axis. The solid purple line indicates precipitation in mm/h, based on IMERG data, as shown in Fig 6b.



Even with a given pathway, the case-study event was extreme, as evident in the properties of each pathway (Fig. 3 and Figure S3). NAlow trajectories are anomalously warm and humid (Fig. 3b,c), and so with a high potential for transporting large moisture volumes (Fig. 3e). As these trajectories descend into the lower troposphere, they undergo adiabatic warming (Fig. 3b) and gradually absorb more humidity (3b). Eventually, they become the most saturated air mass, reaching their peak humidity shortly before arrival into the starting region. Their ascent starts lower in the troposphere and is more rapid than usual (Fig. 3a), leading to a rapid loss of moisture (Fig. 3c). In addition, as in climatology, these trajectories exhibit a potential for the occurrence of convection (Fig. 3f), indicating a higher likelihood of intense precipitation. EAST and WEST trajectories are also warmer, more humid, and show higher equivalent potential temperatures than typically observed, indicating enhanced moisture absorption and buoyancy (Fig. 3e). EAST trajectories, originating in colder and drier conditions, undergo a transformation about 5 days before the event. As they descend into the lower troposphere, they warm up, gain moisture, and increase their relative humidity, contributing significantly to the precipitation in Emilia Romagna between May 15-17, 2023. The percentile ranges and comparative analysis for these properties are further detailed in Figure S4, utilizing boxplots for a visual representation of the data spread for days 0 to -5 from the start of trajectories.

## 6 Predictability and Forecast Interpretation

In the previous sections we have developed a detailed understanding of the dynamics and flow evolution of the May 2023 case study event, and of the broader class of MAM north Italian heavy rainfall events. We have found that the May 2023 event is linked to a chance co-occurrence of high rainfall from multiple airmass pathways, and that each pathway can be linked to a distinct large-scale flow. In Dorrington et al. (2023) it was proposed that flow precursors could be used to improve the predictability of extreme events and understanding of forecast evolution, but with the assumption that a single set of precursors could be used to understand the event. The three-pathway narrative we have described above provides a challenge for such an approach, especially as the operational application of specialised Lagrangian tools for identifying such pathways is impractical, due to the substantial computational resources and extensive time required for calculating trajectories. Therefore in this section we explore how useful flow precursors based on a single-pathway approximation are to understand and raise early signs of the May 2023 event. Figure 12 shows ECMWF 15-day forecasts for 12-hour north Italian rainfall accumulations over the case study period. We show 12 hour accumulation forecasts rather than a 36 hour accumulation in order to highlight the differing evolution for the event onset and peak: forecasts from the 12th onwards contained scenarios with extreme rainfall totals for the earlier target times, with forecast confidence increasing more broadly on the 14th, indicating a 2-4 day skilful forecast range. Forecasts considering rainfall aggregated over only Emilia-Romagna provide the same qualitative conclusion (not shown).

In figure 13 forecasts for daily Z500 and IVTmag lag 0 precursor indices are shown, corresponding to the precursor patterns shown in the top and bottom right of figure 4 and the indices shown in figure 9. As mentioned in the introduction, these precursors were monitored experimentally in late April 2023 onwards, computed by projecting IFS Z500 and IVTmag forecast anomalies onto the reanalysis precursor patterns. From the 8th May, the ensemble forecast predicted somewhat elevated Z500 precursors for the 15th and 16th (i.e. a trough/ridge structure surrounding Italy) in the ensemble mean, but with a wide forecast



spread. By the 10th, the ensemble actually overestimates the trough strength, with what ultimately proves to be an accurate magnitude predicted from the 12th onward.

Forecasts for IVTmag precursors (i.e. strong moisture transport over the central Mediterranean and Italy) for the 15th grow steadily more confident in high values from the 8th onward, with a number of very extreme >$4\sigma$ events in the forecast between the 8th and 10th. For the 16th, forecast IVTmag precursors increase sharply on the 12th, coinciding with the emerging direct

rainfall forecasts. This is due to a time mismatch in earlier forecasts which favoured an extreme from the 14th-15th. The precursor forecasts at long lead times did not provide a perfect prediction of the ultimate extreme, but they did flag clear risks of a possible extreme scenario from the 8th onwards, at a lead time of 7-8 days, which was not visible in the direct rainfall predictions. Such qualitative early signals are of potential utility to the operational forecaster.

Further, the precursors have particular use for understanding very uncertain forecasts. As an example we consider the forecast

from the 8th May, predicting IVTmag for the 15th (a 7 day lead time), indicated by the red box in the top right panel of figure 13. Some members predicted an exceptional moisture transport over Italy, while others indicated a suppressed transport anomaly. To understand the differences between the scenarios captured by this particular ensemble forecast we compute a sensitivity plot: we choose the 5 ensemble members with the highest IVTmag precursor prediction, and the 5 members with the lowest IVTmag precursor prediction, and compute two separate sub-ensemble means. Figure 14 shows the differences in the Z500 flow

field between the two sub-ensembles and clearly indicates that the precise longitude of the Newfoundland cyclone is vital in determining the ultimate downstream development, altering the degree of wavebreaking, and either supporting or suppressing the Westerlies over the Atlas mountains that lead to storm Minerva. Thus we identified a sequence of predictability barriers and identify a clear predictability horizon for the event, suggesting when they will be overcome (after the cyclone's development on the 11th May in the first instance, and after the possible wave-breaking clarifies on the 13th May); information which can

be used by forecasters and by the atmospheric predictability or data-assimilation specialist.

Of course, a natural rejoinder is that we could simply make a sensitivity plot based on members with high/low rainfall over northern Italy, bypassing the concept of flow precursors entirely, as we show in figure 15. However, this simplification can obscure matters instead of clarifying: the actual occurrence and intensity of rainfall is determined by boundary layer and mesoscale details which are both unpredictable on weekly timescales and difficult to reason from, not to mention stochastic

perturbations in the case of the IFS. We see that the overall magnitude of flow anomalies between the two sub-ensembles is smaller than in the IVTmag precursor-based plot, and the source of uncertainty is more ambiguous and spatially diffuse. The actual difference in North Italian rainfall between the precursor-based sub-ensembles is almost as large as the precipitation based sub-ensembles (Supplementary Figures 8 and 9), indicating that this narrative clarity does not come at the cost of overly reduced discrimination. This is not unique to the chosen sensitivity example: Supplementary Section 2 (figures S10 onwards)

presents two additional examples, with very similar results, corresponding to the other red-boxed ensemble forecasts in figure 13.





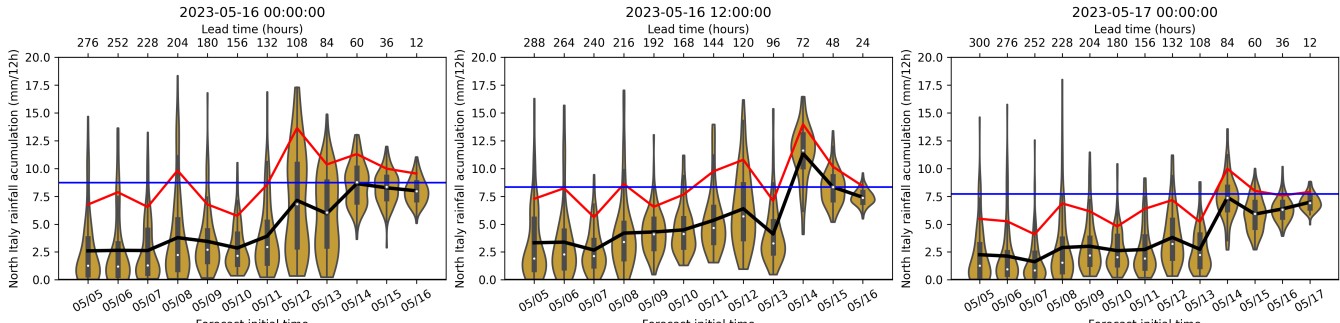

**Figure 12.** Forecast evolution plots showing the ensemble distribution of rainfall accumulations (in mm) over northern Italy in ECMWF medium range forecasts for the 12 hours preceding 16/05/2023 00:00, 16/05/2023 12:00 and 17/05/2023 00:00. Black lines show the ensemble mean, while the box and whisker plots show interquartile range and median values. The red line indicates the 90th percentile of the ensemble. The blue horizontal line indicates the ERA5 estimate of the true outcome.

## 7   Discussion and Conclusions

In this paper we have used two dynamically rooted tools – Lagrangian moisture analysis and event precursor analysis – in order to explore the dynamics and predictability of Springtime north Italian rainfall extremes, including specifically the May 2023 flooding event. While these extremes are of interest and have major societal impact in their own right, we have in large part carried out this analysis in order to demonstrate and emphasise the potential of dynamical thinking for concretely assisting in forecasting extreme events.

The Lagrangian perspective has provided unique and important insights that impact the interpretation and understanding of MAM North Italian rainfall extremes. Firstly, surface moisture sources (accounting for ∼80% of the total moisture content) are primarily local, within the Mediterranean region, with no apparent role for long-range moisture transport. The remaining ∼20% of moisture sources may derive from uptakes beyond the 10-day trajectory analysis we consider, or from processes like horizontal turbulent mixing. Regardless, the airmasses which carry and then precipitate over Italy assuredly have distinct properties depending on their origin, indicating an important non-local pre-conditioning. We have found that these trajectories naturally divide into three origin regions, with low-level recirculating North African airmasses driving the majority of rainfall, due to their high buoyancy and temperature. Airmasses coming from the Atlantic are of secondary importance, while easterly air masses are rare contributors to north Italian rain. In the case study event, from May 15th-17th, all three independent pathways were active and precipitating at different times over the event. From this perspective, the flooding was a dynamically compound extreme. While Lagrangian trajectories are too complex and computationally expensive to be of direct applied use, the insights generated when studying a particular event class have intriguing operational implications. For example, as much of the rainfall over Italy can be attributed to moisture uptake over Italy, this highlights the importance of accurate soil moisture simulation, and the possibility of compounding errors when multiple rainfall events follow each other in swift succession. Further, on the seasonal forecasting (and indeed climate projection) timescale, it is of interest that the dominant







**Figure 13.** Forecast evolution plots as in figure 12 but now for precursor activity indices on 15/05/2023 and 16/05/2023. The blue horizontal line indicates the ERA5 estimate of the true outcome. Red dashed boxes indicate forecasts for which ensemble sensitivity is investigated. As each plot is on a different scale, the y-axis is shaded to aid comparison, and suggests a 'traffic light system' of event risk.



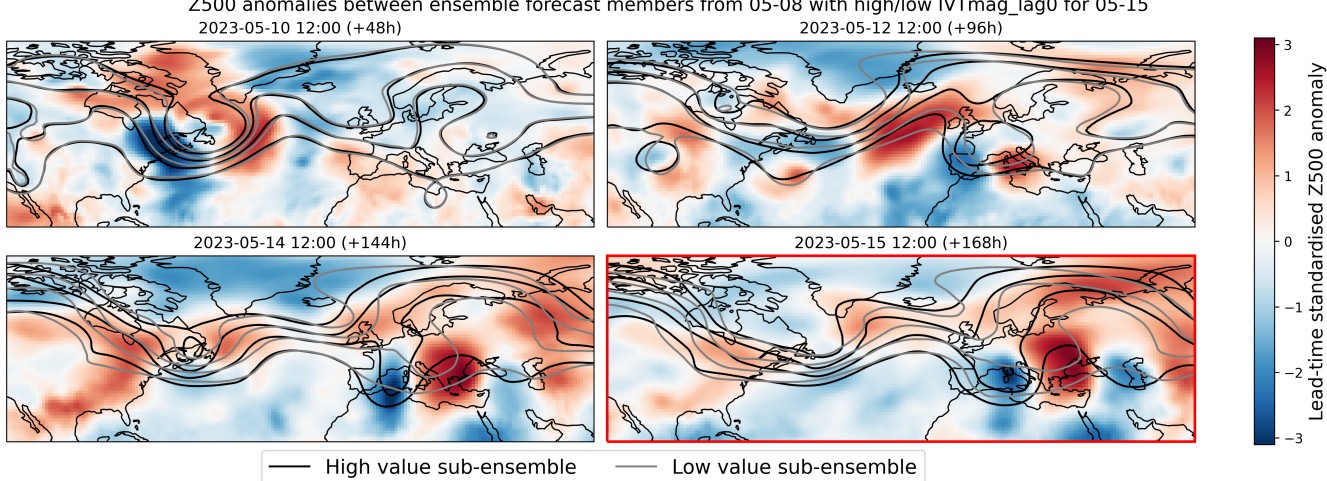

**Figure 14.** An ensemble sensitivity plot showing mean Z500 fields (in contours) for two sub-ensembles of the 08/05/2023 forecast, with their difference (hi-low) shown in shading. sub-ensembles were defined by selecting the 5 ensemble members with the highest/lowest predicted values for the lag 0 IVTmag precursor activity index on 1/05/2023.

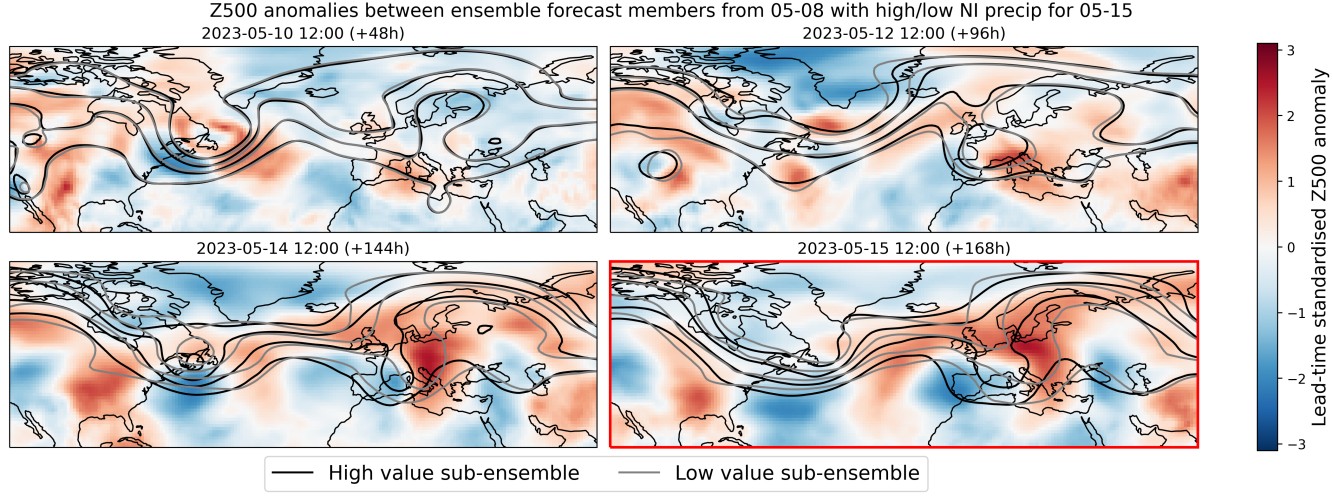

**Figure 15.** As in figure 14 but with sub-ensembles defined by highest/lowest predicted values of North Italian rainfall on 15/05/2023.



NAlow trajectories originate from $\sim$ 35N, and maintain their high potential temperature which helps drive their rapid ascent and heavy rainfall contributions. Seasonal anomalies or forced trends in the meridional temperature gradient may therefore
result in changed severity of north Italian rainfall, although confirming this goes beyond the scope of the current work.

From an event precursor perspective, we are able to reproduce and extend the existing body of knowledge on the large-scale drivers of north Italian rainfall and, crucially, synthesise it into an operationally usable form. The compound nature of the case-study event, contingent on the details of the European wave breaking, paint a complex dynamical picture with limited deterministic predictability, in agreement with the 3 day forecast horizon for direct precipitation seen in the IFS.
However, as the large-scale trough/ridge setup, and strong Mediterranean moisture transport were predicted early on, from a precursor perspective there was early signal 8 days ahead of a possible extreme event, albeit of a qualitative nature. In addition, the precursor perspective allows us to identify the evolution of the earlier Newfoundland cyclone on the 10th May as the discriminating factor between more and less extreme forecast scenarios. After the cyclone develops, strong geopotential height precursors were well predicted, and high IVTmag precursor scenarios emerge, with spread reducing after the downstream
wave-breaking clarifies on the 13th May. Knowing about this predictability barrier *a priori* and when it will be overcome provides guidance on when to expect a more reliable forecast and less spread in the ensemble.

The development of such tools is, in our opinion, timely. Forecasting involves both prediction and interpretation of the weather, but due to the rapidly increasing complexity and data volumes of operational forecasts, with more initialisation times and ensemble members than ever, it becomes more time-consuming for operational forecasters to go beyond the obvious direct-
variable forecasts and to deeply explore the coming scenarios. Therefore synthesis and summary of the relevant large-scale flow features driving an extreme can help to maximise the value of forecasts. Further, we have shown evidence here that precursors can not only flag impactful forecast scenarios, but also help explain them: ensemble sensitivity plots conditioned on precursor activity were more coherent and interpretable than those based on precipitation directly, allowing predictability barriers to be identified.

In contrast to some of the sciences, meteorology is distinct in that the objects of theoretical study are often the exact same ones of relevance to applied practitioners. As we hope to have indicated, there is therefore great potential for dynamical perspectives to improve forecasting directly, and not only through improved process-understanding and model development. The Lagrangian analysis we have performed here has equal potential to inform the understanding of extreme rainfall in any region, while the flow precursors easily extend also to other classes of event, such as wind gusts, heatwaves etc. It is through
the development of such extensible and generalisable frameworks, based ideally on openly available code and datasets, that this potential can be realised. Making our own contribution to this goal, the authors intend to provide proof-of-concept near real-time precursor forecasts, as demonstrated here in the near future, accessible to interested researchers.

*Code and data availability.* Observational and reanalysis data used in this study, from ERA5, IMERG, and ARCIS are all publicly available as described in their relevant citations. The relevant ECMWF historical forecast data is freely available for non-commercial use. Precur-



sors were computed with Domino, an open-source Python package available at `https://github.com/joshdorrington/domino`. Lagrangian pathways were computed with Lagranto-ECMWF, and is freely available as described in the relevant reference.

*Author contributions.* JD and MW carried out data preparation and analysis, and data visualisation with assistance from FG. JD and MW prepared the manuscript with contributions from all authors. All authors contributed to conceptualisation, with supervision from CG.

*Competing interests.* The authors report no competing interests

*Acknowledgements.* The contribution of MW is funded by the German Research Foundation (DFG; Grant GR 5540/2-1) as part of the Swiss-German collaborative project "The role of coherent air streams in shaping the Gulf stream's impact on the large-scale extratropical circulation (GULFimpact)."



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
