# Peer review of "Precursors and pathways: Dynamically informed extreme event forecasting demonstrated on the historic Emilia-Romagna 2023 flood"

_EGUsphere, 2024_

## Author Comment (AC1)

Response to Reviewer Comments

We thank both reviewers for their helpful comments. We detail our response to each individual point below.

Reviewer 1

*The generalizability of certain concluding thoughts, for example about the importance of simulations accurately capturing cyclone development near Newfoundland, could be more firmly established. It is not clear to me whether this identified predictability barrier is applicable also to other northern-Italian heavy-precipitation events, or if the point is simply that further work could be done to determine what sorts of predictability barriers are common across multiple cases.*

Thank you for this comment. We have inserted the following lines in the conclusion to better explain that we expect predictability barriers to be somewhat case-specific:

" We should not expect such predictability barriers to be necessarily generic, although investigating whether systematic predictability barriers emerge when considering a class of extreme events is an interesting avenue for future study. Indeed, the in principle case-by-case nature of such barriers necessitates development of tools to identify them in real-time."

*Both in the aggregate and in this event, there seems to be substantially more moisture uptake from land compared to ocean (Figures 1 and 10). It would be helpful to explain why this is – or maybe why the figures are misleading in this respect -- especially as Section 2 refers to the Mediterranean Sea as the primary moisture source. Relying on previous literature for this would be fine. Is the phenomenon related to convection preferentially occurring over land, perhaps?*

Thank you for your feedback. We recognize the inconsistency in the phrasing as highlighted. To clarify, what we meant was that Mediterranean Sea is the main marine source of moisture, but a large portion of moisture also comes from the surrounding land within the basin. This result is in agreement with previous research by Flaounas et al. (2019) and Gangoiti et al. (2011). The key takeaway here is that most of the moisture originates locally, rather than from distant sources like the North Atlantic. We've revised the relevant sections of the text to ensure these points are communicated clearly and accurately.

*There is much discussion of anomalous moisture and its origins in the Introduction, but I think would be helpful to have more literature review of the role of instability anomalies and/or forced ascent in driving extreme precipitation in Italy.*

Thanks for flagging this oversight. We have now added some discussion of the role of instability and forced assent into the dynamics section:

"These cyclonic systems interact with the steep orography of North Italy, which can induce or intensify convection both through uplift and through forced convergence of the low-level wind (Khodayar et. al. 2021). The combination of orographic forcing and a favourable large-scale flow (i.e. providing a persistent inflow of moist air) can also lead to the development of quasi-stationary mesoscale convective systems, which can lead to extreme and highly localised precipitation (Miglietta et al. 2022). The predictability of, and observational constraints on, these small-scale dynamics are generally poor, and are the reason for the fundamentally probabilistic relation between the large-scale flow and the occurrence of precipitation."

*Title: demonstrated on -> demonstrated for OR demonstrated with*

We believe 'demonstrated on' to be grammatically valid, and carries slightly different meaning to 'demonstrated for' or 'demonstrated with', and so we propose to keep the title as is.

***11****: forecaster's -> forecasters'*

Fixed

***16****: typo*

Fixed

***39****: extreme-events -> extreme events*

Changed

*44-46: a citation or two for this sentence would be good*

*We now provide a citation for each point.*

***47-48****: the commas after 'events' and 'characteristics' should be removed*

Changed

***120****: 'magnitude' would be the more typical term, rather than 'amplitude'.*

Changed

***153****: typo*

Changed

*204: I would think that this potential increased strength of relationship between SST and moisture uptake would have more to do with the types of synoptic weather systems in summer/fall (i.e. more convective, less frontal) than with SST values per se. Or is this perhaps discussed in the Sanchez reference?*

Yes, in Sanchez they explicitly modulate the SSTs and show that this amplifies/suppresses the development of the Medicane they consider. Of course, there is a coupling between the SSTs and the synoptic dynamics, with warmer SSTs increasing boundary layer instability and favouring convective rainfall. The new discussion on instability anomalies in the intro should hopefully help contextualise this.

**212**: *'Dynamical' should be removed, as the sentence refers to both thermodynamical and dynamical characteristics*

Done

*216: Are these negative q tendencies over Italy?*

Yes, the decrease in specific humidity observed in Figure 3c occurs over Northern Italy, predominantly between -12 hours and +12 hours relative to the starting point (0 time) of the trajectories. The figure below (left - case study, right- climatology) shows the average NAlow, WEST, EASTpathways -12h to +12h and the change of specific humidity along them. We observe that the trajectories maintain high humidity levels before reaching Northern Italy, where a significant reduction in humidity occurs. This demonstrates that the observed decreases in humidity are closely associated with our region of interest.

[Figure]

Furthermore, those trajectories by definition are precipitating at the starting location as they are required to experience relative humidity higher than 80% (as defined in the Sodemann et. al. 2008 moisture source detection methodology) and are started from the region at the time when the extreme rainfall has already been identified using ERA5 dataset.

**Fig 2:** *I'm confused about the units here – for comparison with the text, mm/day might be a better choice. The 'May 2023' label at top left should also be moved, perhaps down a bit, to not interfere with the title.*

The label has been moved. The units here are because we are assigning rainfall to trajectories via negative specific humidity tendencies in the Lagrangian trajectories. These cannot be unambiguously translated to mm/day without adding some additional complicating assumptions. However as these results are analysed in a comparative sense, in order to assess the relative contributions of the pathways and the extremity of the May 2023 event, we do not consider this to pose an issue. To better explain, we have altered the figure caption as fpllows:

"Histograms of negative specific humidity tendency (i.e. rainfall) attributable to the (a) NAlow, (b) WEST, and (c) EAST pathways during 66 48 hour extreme rainfall events in northern Italy, computed using Lagrangian analysis. The case study event, from 15th-17th May 2023 is shown with a red dashed line. (d) shows a scatter plot of WEST+EAST vs NALow rainfall totals, indicating their low correlation and that the case study featured strong contributions from all pathways."

*Fig 3: This one is a bit hard to read – I would recommend increasing the line widths. The axis and tick labels are also on the small side.*

The figure has been modified as suggested by the Reviewer.

*253: dependent on -> separated according to*

Changed

*281: Tyrhennian -> Tyrrhenian*

Changed

*282: Appenines -> Apennines*

Changed

*293: While I follow most of this discussion well, the northwesterly flow is hard to see in Fig 5. It might be helpful to add a clarifying remark that it can be seen crossing France, then plunging south into Algeria and back to Italy, at least on the 15$^{th}$.*

Thanks for this suggestion, we now have expanded the sentence as follows: "The remnant east-Atlantic ridge, widening and decaying by the 16th, also supports North-Westerly flow around the Alps on the 15th and 16th, which can be seen crossing France and plunging into Algeria, before recirculating into Italy. At the same time, the anticyclonic anomaly over eastern Europe supports easterly flow into the central Mediterranean."

*Fig 7: The labeling of this figure needs improvement in image quality and in the text*

Done.

*Fig 8: Line 281 states Storm Minerva was located in the Tyrrhenian Sea, while here the Adriatic is mentioned for May 16. The geopotential map would seem to support the Tyrrhenian, however – unless these phrases refer to different days?*

As Minerva's spatial extent is large enough to touch both Italian coasts when it makes landfall, and in different 6-hourly periods could be said to be in either sea, we have recaptioned this subplot "Minerva reaches Italy".

*313: I don't see this – Fig 3e looks to show that theta-E is highest for NAlow trajectories (and that East trajectories have only slightly higher values)?*

Thanks for capturing this unclear point, this was intended to be a West-East comparison, which we now make clear:

"This is further supported by the subsequent analysis, which reveals that EAST trajectories also maintain higher levels of equivalent potential temperature than WEST trajectories"

**320**: *'Particularly' can be removed as redundant. 'Unusual' might also be a better choice than 'unique'.*

Changed to "dynamically unusual"

**330**: *It might not be necessary to add, but on this point for me, Fig 7 helped to illustrate that the low-level flow almost perfectly circles around the Italian peninsula without encountering major topographic barriers before reaching Emilia-Romagna.*

Thank you for this helpful suggestion. We now include the below sentence:

"The resulting structure of the low level flow (c.f. figure 7 circles round the topographic barriers of central and southern Italy which might otherwise trigger precipitation, before reaching Emiglia Romagna from the north Adriatic"

**Fig** *11: It could be made clearer in the caption and/or the main text that (if my interpretation is correct) this figure compares inferred precipitation from the trajectory analysis and observed precipitation from satellite data.*

This interpretation is indeed correct, and we now say " The bars, divided into pathways from Fig.10, represent the total moisture loss inferred from trajectory analysis in kg/day for the dates specified on the x-axis. The solid purple line indicates precipitation in mm/h, based on IMERG satellite data, as shown in Fig 6b."

**335**: *It's unclear what 'most saturated' means in this context.*

Changed to 'most humid'

**348**: *The word 'chance' is confusing here; I think it could simply be removed without much loss of meaning.*

Removed

**373**: *A citation that discusses this potential utility in some way would be helpful, as the point is not immediately evident to me (i.e. perhaps many forecasts in general have a small number of ensemble members showing extreme cases that never come to pass?).*

Yes, as the reviewer rightly points out, you would typically expect e.g. 1 member to show a 99th% extreme in a hundred member ensemble. But if 5 members predict such an extreme, this is now a situation worthy of early monitoring. The text has been amended with:

"Such qualitative early signals are of potential utility to the operational forecaster, provided the forecast is reliable. While large ensemble forecasts will often include extreme scenarios, understanding at a glance when extreme risk is elevated/suppressed provides a useful operating heuristic."

In a paper currently in preparation we confirm that the precursors are predicted reliably in the IFS and provide well calibrated probability estimates of extremes.

**405**: *no -> little*

Changed

**407**: *carry moisture?*

Fixed

**412**: *It would be more precise to say 'contributing to precipitation'.*

Changed

*420: Are there any studies that suggest this in the Mediterranean broadly, for example?*

To our knowledge there is nothing written on this topic which lies at the intersection of Lagrangian analysis, extreme hydrology and climate change research. However, a preliminary analysis shown below points towards the feasibility of this hypothesis:

[Figure]

[Figure]

[Figure]

[Figure]

Monthly TP anomalies conditioned on monthly $\frac{dT}{dx}$, averaged over 5-35W, 20-40N

[Standardised anomaly composites of monthly MAM ERA5 total precipitation between 1940 and 2023, conditional on terciles of the monthly 800 hPa meridional temperature gradient between 20-40N, averaged over 5-35W as shown by the red box].

In essence this gradient provides a first approximation of how anomalously warm a low-level North African air mass advected north into the mediterranean will be. We see that when this gradient is strong, rainfall anomalies in the adriatic are high.

**446-447**: *I am a little confused by the wording of this sentence.*

We have rephrased as follows:

" In our own ongoing work, the authors intend to use the approach demonstrated here to produce real-time precipitation precursor forecasts, as a proof of concept application accessible to interested researchers."

Reviewer 2

*The study and the methodology applies to large-scale and relatively long-duration precipitation events (as opposed to convective extremes that are examined in other studies). I think this should be stated more clearly in the abstract and introduction.*

We now clarify that we consider "48-hourly extreme rainfall" in the abstract, and explicitly state " Our focus here is on larger-scale organised rainfall events with multi-day persistence." in the introduction.

**Line 26** – *I don't understand what you mean by "increased rainfall probability" here*

To clarify, the sentence now reads " Skilful probabilistic predictions of rainfall occurrence rarely exceed a week ahead"

**Line 91**: *both is a repetition*

Removed

**Line 118**: *"defining an event as a day with rainfall exceeding the 90th percentile of this index (≈ 8.5mm/day)" is this a spatial average over the domain? The peak? Please specify*

Replaced "index" with "domain average" to clarify.

*Line 141: why up to 480 hPa?*

This follows exactly the methodology of Sodemann et al 2008, which allows us to compare our results directly with prior literature. In practice, as we also have a minimum specific humidity requirement, the results are mostly invariant to the bounding altitude as the middle-to-upper atmosphere is very dry. An earlier iteration of the work with a 700hPa ceiling gave almost exactly the same trajectories.

**Line 151**: *"lower" instead of "smaller"?*

*Lines 156-158: this explains why 5 days are used for NA but not why 7 days are used for the other categories.*

Firstly, there was a typo: 7 days were used for NA and 5 for West and East. We now explain this difference in the text. When using a 7-day threshold, some trajectories approaching Italy from the West are categorised as EAST as they have circumnavigated the globe in the previous week. A 5 day threshold avoids this complication.

**Line 163**: *I think the t in qt should be subscript*

Fixed.